# Tomato Defense against Whiteflies under Drought Stress: Non-Additive Effects and Cultivar-Specific Responses

**DOI:** 10.3390/plants11081049

**Published:** 2022-04-12

**Authors:** Francisca J. González-Klenner, Marta V. Albornoz, Germán Ávila-Sákar, Jaime A. Verdugo

**Affiliations:** 1Escuela de Agronomía, Facultad de Ciencias Agronómicas y de los Alimentos, Pontificia Universidad Católica de Valparaíso, Quillota 2260000, Chile; francisca.janine@gmail.com; 2Centro Regional de Investigación e Innovación para la Sostenibilidad de la Agricultura y los Territorios Rurales, Ceres, Pontificia Universidad Católica de Valparaíso, Quillota 2260000, Chile; malbornoz@centroceres.cl; 3Department of Biology, University of Winnipeg, Winnipeg, MB R3B 2G3, Canada; 4School of Pedagogies in Natural and Exact Sciences, Faculty of Education Sciences, University of Talca, Linares 3580000, Chile

**Keywords:** defense against herbivores, greenhouse whitefly, insect–plant interaction, resistance, *Solanum lycopersicum*, *Trialeurodes vaporariorum*, tolerance, trichomes

## Abstract

Two of the main causes of losses in tomato production are the greenhouse whitefly, *Trialeurodes vaporariorum* (Hemiptera: Aleyrodidae), and drought, which is becoming a central problem in agriculture due to global climate change. The separate effects of whitefly infestation and drought have been amply studied in many crop systems. However, less is known about their combined effects. To evaluate whether drought stress (DS) affects plant defense against whiteflies, we assessed the joint effects of whitefly infestation and DS on plant vegetative and reproductive performance in four tomato cultivars, and assessed the effects of DS on plant resistance and tolerance (compensatory ability) to whiteflies in a greenhouse experiment. Generally, we found negative effects of DS and whiteflies on plant performance, but the combined effects of DS and herbivory were not worse than those of either stress alone. In fact, plant performance under the combined effect of both stresses was usually similar to that in the presence of whiteflies without DS. Plants growing under DS had greater trichome density. However, plant resistance—as measured by whitefly population growth—decreased under DS in two cultivars and was unaffected in the other two. Compensatory ability decreased under DS in all but one cultivar. These cultivar-specific responses suggest genetic variation in resistance and tolerance to whiteflies and could be associated with differences in drought tolerance among cultivars. Our findings underscore the difficulty in predicting the combined effects of DS and herbivory and point to the need for a better understanding of the mechanisms underlying plant responses to both stresses at the molecular, cellular, and organismal levels.

## 1. Introduction

Plants in managed and wild populations are simultaneously responding to the biotic and abiotic components of their environment. Drought and herbivory—two such components commonly faced by plants of many habitats—and their combined effects on plants are expected to become exacerbated by climate change [1,2,3]. Both factors are known to have detrimental effects on plant growth and reproductive output, albeit mostly from studies that address each stressful factor singly [4,5,6,7] and refs. therein.

Predicting the simultaneous effects of drought stress (DS) and herbivory on plant growth and reproduction is not straightforward [8,9]. Several hypotheses have been proposed to address how DS affects plant responses to herbivores. According to the Plant Stress Hypothesis [10,11], DS plants should be less resistant than well-irrigated plants because water deficit increases tissue nitrogen concentration, making tissues more nutritious, and therefore more attractive to herbivores. However, plants under prolonged or constant DS might lose turgor pressure to the point that they become less suitable for herbivores, particularly phloem feeders [12,13]. Effectively, this would make plants under DS more resistant to herbivores. 

The same prediction is derived from the Plant Vigor Hypothesis [14], according to which vigorous plants are more suitable hosts for herbivores (presumably because plants that grow more or faster constitute a supply of greater diversity and quantity of the nutrients needed by herbivores than non-vigorous plants). Since DS plants are not vigorous, they are expected to be more resistant to herbivores than well-watered (WW) plants. It must be noted that a vigorous plant could also have greater access to resources needed for the production of defensive compounds, which would reverse the direction of this prediction.

The Plant Growth-Differentiation Balance Hypothesis (summarized in [15]) posits that plant investment in defense occurs maximally under conditions that favor intermediate growth rates, because under conditions that severely restrict growth, photosynthesis becomes so limited that plants cannot allocate (carbon-based) compounds to defense. However, under conditions that favor fast growth, plants allocate little or no resources to defense because resources are allocated to growth. Therefore, moderate DS should result in greater resistance to herbivores, while severe DS would make plants less resistant than those under adequate watering. The main difficulty in testing this hypothesis is establishing a priori what “moderate drought stress” is for different species in terms of the effect of drought on plant growth rate.

Notably, these hypotheses consider plant defense in terms of resistance, i.e., the ability of plants to avoid being damaged by herbivores. However, plant defense against herbivores may also occur by means of tolerance: traits that allow plants to maintain fitness despite being damaged by herbivores [16,17]. Given the potential trade-off between resistance and tolerance [18], if resistance to herbivores decreases under drought, the reverse could occur with tolerance to herbivores (see also [19]).

In terms of the biochemical pathways involved in stress responses, drought is known to cause increased endogenous levels of abscisic acid (ABA), a phytohormone that, among other effects, inhibits the salicylic acid (SA) pathway [20,21], which, in turn, is crucial in plant defensive responses to pathogens and phloem feeders. Consequently, plants should become less resistant to phloem feeders under DS [21] —but see [22]. However, DS is also known to induce greater trichome density on leaves, which could protect leaves simultaneously against desiccation [23,24,25]—but see [26]—and against herbivores [27,28] and refs. therein. In tomato (*Solanum lycopersicum* L.) and its close relatives, eight trichome types have been described [29]. Four of these are glandular (types I, IV, VI, and VII) and four are non-glandular. These trichomes have been shown to confer resistance against whiteflies and chewing herbivores, and can also help plants to tolerate drought and cold [28,30,31].

Tomato is one of the most important crops worldwide [32], and the greenhouse whitefly, *Trialeurodes vaporariorum* Westwood (Hemiptera: Aleyrodidae), is one of the crop’s most prevalent pests, and among the main causes of severe economic losses to the tomato crop industry [33]. The polyphagous and cosmopolitan *T. vaporariorum* may have detrimental effects on plant growth and reproduction either directly or indirectly. Direct effects result from the extraction of phloem sap rich in carbohydrates, amino acids, water, and other resources needed by plants for tissue maintenance, growth, and reproduction [34]. Indirect effects result from whitefly-mediated diseases such as sooty mold: the colonization by *Cladosporium* molds of leaves and fruits coated with whitefly honeydew. As sooty mold covers leaves, photosynthesis decreases, with consequent declines in carbon and nutrient acquisition [35,36]. Sooty mold on fruits needs to be washed off for marketing [37], which adds to the production costs for tomato producers [38]. Whiteflies also mediate viral diseases, which generate great economic losses mostly through direct damage to the plants [39,40,41,42]. 

In this study, we used a tomato–whitefly plant–herbivore system to test whether plants under DS produced leaves with greater trichome density, and whether this resulted in greater resistance against whiteflies. We assessed resistance in terms of trichome density and whitefly performance (population growth rate) on plants of four cultivars purported to vary for resistance to whiteflies [43]. We also assessed the compensatory ability of plants to herbivore damage (a proxy of tolerance of herbivory [38]) to test the proposed trade-off between resistance and tolerance [18,44]. 

## 2. Results

### 2.1. Trichome Density

Only 21 plants had Type I glandular trichomes, and 19 of these were DS plants. In the WW treatment, only two Luciana plants had this kind of trichome (irrigation effect: *χ*^2^ = 16.56, d.f. = 1, and *p* < 0.0001; Figure 1; see also Appendix A). The density of type VI glandular trichomes was greater in DS than for WW plants (*χ*^2^ = 1122.9; d.f. = 1; *p* < 0.0001), but the magnitude of the difference varied by cultivar (cultivar–irrigation interaction effect: *χ*^2^ = 90.1; d.f. = 3; *p* = 0.0010; eight-fold in Mistral, three-fold in Patrón; Figure 2). Among DS plants, cultivar Seven had the greatest type VI trichome density of all.

As with glandular trichomes, the density of non-glandular trichomes (type V) was greater under DS (*χ*^2^ = 4287.3; d.f. = 1, *p* < 0. 0001), except for Mistral, in which type V trichome density was lower under DS primarily because Mistral plants had greater trichome density under WW conditions than the other cultivars (cultivar–irrigation interaction effect: *χ*^2^ = 2271.5; d.f. = 3, *p* < 0. 0001; Figure 3).

### 2.2. Whitefly Performance

The population growth rate of whiteflies was generally positive, except for whitefly populations on DS Mistral and Patrón plants, which had a negative population growth rate (cultivar–irrigation interaction effect: *F*_1,112_ = 20.93; *p* < 0.0001; Figure 4).

### 2.3. Plant Performance

We found significant cultivar–irrigation–whitefly interaction effects on plant height, stem diameter, and number of leaves produced (Table 1). Under WW conditions, there was a clear negative effect of whiteflies on all three variables. A negative effect of DS was evident on plant height and number of leaves, but not for stem diameter. Interestingly, the negative effect of whiteflies on plant height under DS was only evident in Mistral plants. We found no effect of DS on the stem diameter, and the negative effect of whiteflies was independent of DS, except in Luciana plants, where we found no whitefly feeding effect under DS. Except for Luciana plants under DS, whitefly feeding had a negative effect on the number of leaves produced (Figure 5). 

Given the potential correlation between plant height, diameter, and number of leaves, we conducted a Principal Component Analysis (PCA) and retained for further analyses Principal Component 1 (PC1), which accounted for 0.74 of the variance and had loadings of 0.61, 0.48, and 0.63, respectively, for each of the aforementioned variables. Using PC1 as an overall measure of vegetative performance in a generalized linear model, we found negative effects of DS (*F*_1,240_ = 23.3; *p* < 0.0001) and whiteflies (*F*_1,240_ = 156.4; *p* < 0.0001), but these varied by cultivar (cultivar–irrigation–whiteflies interaction: *F*_3,240_ = 3.85; *p* = 0.010): whiteflies generally had a more negative effect on vegetative performance than drought, except in Luciana, where DS and whiteflies had negative effects of a similar magnitude (Figure 6). Interestingly, DS did not significantly impair the vegetative performance of Mistral plants.

### 2.4. Flowering

We found a significant cultivar–irrigation interaction effect on the days to first flower (*χ*^2^ = 50.1; d.f. = 3; *p* < 0.0001; Figure 7): DS plants of three cultivars tended to bloom earlier than WW plants, but the difference between irrigation treatments was only significant in Mistral, in which DS plants flowered nine days earlier than their WW counterparts. Differences among cultivars were only significant under DS: Luciana took longer to flower than the other cultivars.

### 2.5. Fruit Production and Fruit Biomass

We found a negative effect of whitefly feeding on the number of fruits produced per plant (*χ*^2^ = 173.91, d.f. = 3, and *p* < 0.0001; Figure 8A). However, the effect was much more pronounced in Mistral plants (cultivar–whitefly interaction: *χ*^2^ = 16.26, d.f. = 3, and *p* = 0.001), and under WW conditions (irrigation–whitefly interaction: *χ*^2^ = 7.29, d.f. = 1, and *p* = 0.007). Interestingly, fruit production was lowest under combined DS and whitefly feeding. Fruit fresh weight also tended to be negatively affected by whitefly feeding, but the magnitude and significance of the effect depended on both cultivar and irrigation (cultivar–irrigation–whitefly interaction: *F*_3,192_ = 3.98 and *p* = 0.009; Figure 8B). Under WW conditions, the negative effect of whitefly feeding on fruit fresh weight was significant only in Luciana, and under DS, the same was true only in Mistral. Indeed, in Mistral, the fruits with the lowest fresh weights were produced under combined DS and whitefly feeding. DS alone also had a negative effect on fruit fresh weight, but this was only marginally significant in Luciana. In Seven, fruit fresh weight was greater on WW plants regardless of whitefly presence (Figure 8B). Fruit dry matter content was generally greater in DS than WW plants (*F*_1,192_ = 62.71; *p* < 0.0001; Figure 8C). Fruits of Luciana had lower dry matter content than those of the other cultivars (*F*_3,192_ = 3.96; *p* = 0.009).

### 2.6. Compensatory Ability

We found significant effects of DS (*F*_1,152_ = 8.42; *p* = 0.004) and cultivar (*F*_3,152_ = 2.84; *p* = 0.040), but also of their interaction (*F*_3,152_ = 2.893; *p* = 0.037) on compensatory ability: compensatory ability decreased with DS in three cultivars, but not in Luciana (Figure 9). Contrary to the general trend, Luciana plants tended to compensate better under DS. Under WW conditions, plants of cultivar Seven were the best at compensating for whitefly damage, while under DS, Luciana plants were the best at compensating.

## 3. Discussion

Several studies have separately assessed the effects of drought and whitefly infestation on growth and fruit or seed production in tomato and other crops [4,5,6,7] and refs. therein. Here, we used the tomato–whitefly plant–phloem feeding herbivore system to study how plants respond simultaneously to drought and herbivory. Attention to the combined effects of these factors has become particularly relevant in the light of global climate change, which has already brought greater drought duration and severity in some geographic areas [45], and is bound to alter plant–insect interactions in many ecosystems. Moreover, plant responses to several kinds of stresses, including drought and herbivory, occur through elicitation of biochemical pathways that partially overlap [20,21]. For this reason, exposure to one stress factor could prime plants to respond to another one [46].

In this study, we generally found that the effects of DS and whitefly herbivory on plant performance were not additive. While both kinds of stress had detrimental effects on the variables measured, generally, the combined effects of DS and herbivory were not worse than those of either stress alone. In fact, plant performance usually decreased more in response to whiteflies than to DS, and the level of performance under the combined effect of whiteflies and DS was similar to that in the presence of whiteflies without DS. While this result suggests greater tolerance of herbivory under DS, our analysis of compensatory ability did not confirm that pattern (see below). 

### 3.1. Effects of DS on Resistance to Herbivores 

We had hypothesized that increased trichome density induced by DS could provide plants with greater resistance against herbivores [31]. In general, we found greater density of all three types of trichomes for plants grown under DS, although we did not see this pattern for Type I trichomes in Luciana, or for non-glandular trichomes in Mistral. Increased trichome density in response to DS has been observed in olives (*Olea europaea*) and *Lotus creticus*, among other species [47,48,49,50], as well as in tomato, but only under low-light conditions [51]. Indeed, trichome density has been shown to be positively associated with water use efficiency in tomato [25], although it may not change in response to water deficit in drought-adapted cultivars [48,52], which is what we saw in Luciana for Type I trichomes and Mistral for non-glandular trichomes. 

Together, our findings of greater trichome density under DS and negative whitefly population growth rates only under DS, in two cultivars (Mistral and Patrón) suggest that plants under DS become more resistant to whiteflies than WW plants, albeit this change may be cultivar-specific. Greater resistance against whiteflies under DS is contrary to White’s [10] Plant Stress hypothesis prediction, but consistent with the predictions of Hubberty and Denno’s [12] constant stress and loss of turgor hypothesis, and Price’s [14] Plant Vigor hypothesis. 

Also, consistently with our finding of negative whitefly population growth rates on DS plants, whiteflies of a different species (*Bemisia tabaci*) were found to avoid DS tomato plants [53]. However, that same study found the leaf miner, *Tuta absoluta*, to have the opposite preference pattern (and thus, lower resistance of DS plants to this herbivore). Yet another study found no effects of DS on *B. tabaci* performance on two other tomato cultivars [54]. In cotton, *B. tabaci* has been found to be more attracted to DS plants [55], and to have improved colonization and oviposition on water-stressed plants [56]. In contrast, *B. argentifolii* was found to have reduced oviposition on DS cotton in greenhouse and field experiments, albeit with some variation among experiments [57]. Thus, the effect of DS on resistance seems to be both herbivore- and host-plant-specific. 

While we did not measure ABA levels, our finding of greater resistance under DS (in two cultivars) contradicts the lower resistance predicted on the basis of the known elicitation of ABA by DS, and the consequent suppression of the SA pathway by increased ABA levels [21,58]. In contrast to the jasmonic acid-based response of plants to chewing herbivores, plant resistance to sap feeders usually involves the SA pathway. Therefore, the suppression of the SA pathway would make plants more susceptible to whiteflies. Given our finding of greater (or unchanged) resistance to whiteflies under DS, we propound that other traits that change in response to drought must be driving the increase in resistance against whiteflies of DS plants of those cultivars that show such a response. One such trait could be trichome density.

Given the importance of glandular and non-glandular trichomes in plant resistance to insect attacks [59,60], increased trichome density induced by DS might have interfered with whitefly feeding, which, in turn, could drive the reduction of the negative effects of whiteflies on growth and fruit production observed for most cultivars in our study. Type I trichomes, common in several wild species of tomato, are the main sites of synthesis of a variety of acyl sugars, which are allelochemicals that have negative effects on arthropod pest performance [29,61]. The density of Type I trichomes may be an important determinant of the responses seen in our experiments, especially considering that the lack of differences between DS and WW Luciana plants is consistent with the lack of response to DS in Type I trichome density seen in this cultivar. Moreover, Mistral and Patrón had the highest Type I trichome density under DS, and these were the two cultivars in which whitefly population growth rate was negative under DS, thus highlighting the importance of Type I trichomes as a resistance trait against whiteflies. Importantly, plants under DS showed greater density of Type VI trichomes, which are also involved in metabolite storage in *S. lycopersicum* and one of its close wild relatives, *S. habrochaites* [62].

We speculate that apart from changes in trichome density, there are other species- and cultivar-specific metabolic changes brought about by DS in tomato that affect whitefly feeding, nutrition, growth, and reproduction. For instance, some of the different plant responses to *Trialeurodes* and *Bemisia* could be due to a greater detoxification capacity of *Bemisia*, most probably co-opted from plants via horizontal gene transfer [63]. 

### 3.2. Effects of DS on Compensatory Ability

Based on the predicted trade-off between resistance and tolerance due to their potentially high direct costs and purported overlap in defensive function [18], we had hypothesized a lower tolerance against herbivores in DS plants. While compensatory ability seemed unaffected by DS in two cultivars, the other two (Mistral and Seven) had lower compensatory ability under DS, as predicted. 

### 3.3. Plant Vegetative and Reproductive Performance

In general, we found a negative effect of DS on plant vegetative performance for plants without whiteflies, but not for Mistral, which suggests a certain degree of drought tolerance of this cultivar. In contrast, Luciana plants suffered greater negative effects of drought on several variables compared to the other cultivars, which could suggest that Luciana plants were less tolerant of drought. 

Fruit production tended to decrease with either drought or whitefly infestation, but the responses varied by cultivar. In general, whiteflies caused a drop in fruit production similar in magnitude to the one caused by drought, except in Mistral, for which the drop was greater in response to whiteflies. In Luciana, the combination of DS and whitefly infestation brought fruit production lower than infestation alone. The lower fruit production of Luciana plants may be what allowed this cultivar to compensate better (be more tolerant of whiteflies) under DS: it does not take much for plants under whitefly attack to produce as many fruits as plants without whiteflies.

DS may affect the number and size of fruits produced through the retardation of growth that results in smaller plants. Smaller plants, in turn, will have fewer floral meristems and a reduced ability to provision fruits and bring them to maturity [64]. In addition, given that water makes 90–95% of the volume of a fruit (Figure 8C), lack of water must be directly limiting the growth of fruit tissues. This would explain the decrease in fruit fresh weight seen in all cultivars in our study, which agrees with earlier findings in DS tomato [65].

DS also initiates a series of changes in the hormonal network of plants that may result in the induction of precocious flowering, but also of flower and fruit abortion [66,67,68]. Premature flowering, as found for two of the cultivars we studied, and the production of smaller fruits, are well known responses to DS in tomato (e.g., [69]), but they may depend on the timing of drought with respect to flower and fruit development [66]. 

## 4. Materials and Methods

### 4.1. Experimental Design

To assess the effects of DS on plant resistance and tolerance to whitefly feeding, we grew tomato plants of four commercial cultivars (Luciana, Mistral, Patrón, and 7742, which we dubbed “Seven” for brevity and convenience in data management) under contrasting conditions of water availability (WW and DS), with and without whiteflies. We used four cultivars that varied in resistance to whiteflies [38] and possibly also in tolerance to drought. Plants were obtained from a commercial nursery, transplanted into 2.5 L plastic pots containing sterilized leaf mulch substrate in a greenhouse, and watered sufficiently during the following three weeks to promote their establishment. Plants were not subjected to fertilization, to avoid potential changes in resistance due to mineral content [38]. 

Initially, 480 seedlings were established (120 plants per cultivar); 120 plants were used to assess resistance based on whitefly population growth (insect performance). Of the remaining 360 plants, we took 320 to split equally between whitefly treatments (160 with/160 without). The rest were used to measure water potential (destructive sampling). However, we later realized we needed more plants for water potential measurements to be representative of the two irrigation treatments in each cultivar, so we used 56 plants without whiteflies for these measurements. This left us with 104 plants without whiteflies, eight of which died during the study. Thus, in total, we had 160 plants with whiteflies (80 WW; 80 DS), and 96 without whiteflies (46 WW; 50 DS). 

A greenhouse was especially built within the lands of the Pontificia Universidad Católica de Valparaíso in Quillota, Chile, using anti-aphid mesh (3330 Biorete BT 16/10, Arrigoni Spa, Quilicura, Chile) to preclude the unwanted access of whiteflies (and other pests). The greenhouse consisted of two chambers separated by anti-aphid mesh to keep apart plants of different whitefly treatments (with/without). Entrance to each chamber was through a vestibule to prevent the undesired entrance (or exit) of whiteflies that could alter the intended treatments. Chambers were split into 16 cubicles: two per cultivar. Plants were assigned to cubicles according to their cultivar so as to avoid potential discrepancies in the number of whiteflies per plant, due to differences in cultivar attractiveness to whiteflies (10 plants per cubicle for the chamber with whiteflies; 6 plants per cubicle for the chamber without whiteflies). Cubicles were randomly assigned to each cultivar to minimize positional effects inside the greenhouse (see whitefly addition treatment below). 

Contrasting water availability treatments were achieved using an automatic irrigation controller (Orbit^®^ 57894 Easy Set Logic, North Salt Lake, UT, USA) connected to an irrigation system with self-compensating drippers (irrigation emitters), with one of the following flow rates for each pot: 2 L·h^−1^ for 25% of field capacity (DS irrigation treatment), and 8 L·h^−1^ for 100% field capacity (WW irrigation treatment). The irrigation uniformity was tested according to the methodology of Villavicencio and Villablanca [70]. WW plants were watered daily for one minute and received a total of 133 mL·day^−1^, while plants under the low irrigation regime were watered every second day for a minute and received 33 mL·day^−1^. Plant water potential was measured with a Scholander pump (Pump-Up Chamber, PMS Instrument Company, Albany, OR, USA) to determine when plants became drought-stressed [71]. Water potentials between −0.4 and −0.6 MPa were considered optimum irrigation (WW plants); plants with water potential values of less than −0.9 MPa were considered to have DS [34]. Water potential was measured on plants at various locations within the greenhouse to avoid positional bias. Because the measurement of water potential is destructive, we only took measurements of about 20% of plants in each treatment.

Whiteflies were reared for five generations in a separate polycarbonate greenhouse on tomato plants of a different cultivar (Afamia) to avoid whitefly adaptation to any of the cultivars used in the experiment. Whitefly-infested leaves of these plants were collected and transferred in bags for their release within the corresponding cubicle of the main greenhouse. We released 220 individuals in each cubicle (roughly 22 individuals per plant). Insects were placed on plants 60 days after transplantation, when plants in the DS treatment exhibited signals of DS and had leaf water potentials lower than –0.9 MPa. The duration of the trial was 121 days from seedling transplant to the harvest of mature fruits (ideal harvest time in the study area), except for the insect performance trial, which took less time.

#### 4.1.1. Trichome Density

To assess the influence of DS on trichome density as a putative mechanism of resistance [28,51,62], we identified and counted glandular (types I and VI) and non-glandular (type V) trichomes on three 1-cm^2^ pieces taken from each of two leaves per plant–cultivar–irrigation treatment. Density measurements of both leaves were averaged for each plant. We used leaves from the upper third of the plant, where the adult whiteflies are found in greater abundance. Observations were carried out with a compound microscope (BA 210, Motic, Xiamen, China) connected to a digital camera (Moticam 3, Motic, Xiamen, China). Trichomes were classified, counted, and recorded as per McDowell et al. [29]. 

#### 4.1.2. Whitefly Performance

One leaf of the upper third of each plant was infested with ten female individuals of *T. vaporariorum* enclosed in a tulle bag to prevent them from escaping. Five plants were used for each combination of cultivar and irrigation. The plants used in this test had the two irrigation levels, but they had no contact with whiteflies previously to avoid a defensive response of the plant. After ten days, eggs, nymphs and adults were counted. The population growth rate (PGR) was estimated as $\mathrm{PGR}=\left(\ln\left(N_{2}\right)-\ln\left(N_{1}\right)\right)/\left(t_{2}-t_{1}\right)$, where *N_*1*_* is the initial population, *N_*2*_* final population, and $\left(t_{2}-t_{1}\right)$ is the number of days of the trial (ten days) [72].

#### 4.1.3. Plant Vegetative and Reproductive Performance

To assess the influence of irrigation and whitefly infestation on plant vegetative and reproductive performance (including compensatory ability), we measured height, diameter of the main stem, number of leaves and fruits produced, and fruit fresh weight and dry matter content. We also recorded the date of anthesis of the first flower, which generally occurred before we introduced the whiteflies to the appropriate greenhouse chamber. Height was measured from the rim of the pot to the uppermost leaf axil on a plant. Diameter was measured 10 cm above the top edge of the pot with a digital caliper. The total number of leaves produced by a plant was obtained by cumulative counts of leaves until the onset of senescence. All fruits on a plant were counted at the time of maturation of the first fruit (a commercially mature fruit is completely red; grade 5 or 6 of López Camelo [73]). Three commercially mature fruits were harvested from each plant (107–121 days after plants were transplanted to pots) and weighed. The average fresh weight per individual was used for statistical analyses. Subsequently, fruits were chopped into small pieces, re-weighed to obtain the fresh weight of the pieces, and dried in an oven (EST 200, Amilab, Santiago, Chile) at 80 °C for 22 h, to reach a constant weight [74]. Fruit dry matter content (DMC) was calculated as $DMC=100\left(M_{f}/M_{i}\right)$ where *M_i_*, and *M_f_* are, respectively, the initial and final sample weights, and *DMC* is expressed as a percentage [75].

#### 4.1.4. Compensatory Ability

Tolerance is a reaction norm that can be estimated as the slope of the function between fitness and the amount of tissue removed by herbivores. However, this requires the use of closely related individuals and usually results in low statistical power to detect slopes different from zero. To avoid these limitations, we used compensatory ability, a measurement at the individual plant level akin to tolerance. We estimated compensatory ability as
$$C_{ijd}=\frac{F_{ijd_{infested}}}{{\tilde{F}}_{jd_{uninfested}}}-1$$
where *C_ij_* and $F_{ijd_{infested}}$ are, respectively, the compensatory ability and the number of fruits produced by individual *i* of variety *j* grown in irrigation treatment *d*, and ${\tilde{F}}_{jd_{uninfested}}$ is the median of the number of fruits produced by individuals of variety *j* in irrigation treatment *d*. Thus, values of *C_ij_* greater, equal, or lower than zero indicate, respectively, over-, equal- (or exact-), or under-compensation to whitefly feeding [38].

### 4.2. Statistical Analysis

We evaluated the effect of DS on trichome density, whitefly population growth rate and days to first flower by means of generalized linear models (GLM) using a Poisson distribution. We also used GLMs to evaluate the effects of irrigation and whitefly feeding on height, diameter, number of leaves, number of fruits, fruit weight, and fruit DMC and compensatory ability of plants. For balanced designs, we used sequential sums of squares (SS) and always entered the cultivar before the irrigation treatment effect. For unbalanced designs, we used Type II SS see [76,77]. Type I trichome counts were generally zero (459 of 480 counts). Therefore, we transformed that variable into a presence/absence binomial variable and used a GLM with a binomial error and logit link to test the effects of cultivar and irrigation on the presence of this type of trichome. Whitefly performance and those parameters with normal distribution were analyzed using a factorial ANOVA followed by a Tukey HSD multiple comparisons test. Statistical analyses were performed using R packages stat, car, and gof [78].

## 5. Conclusions

This study has shown that the effects of whiteflies and DS on tomato are generally negative and not additive (performance was not necessarily worse under the combined effects of DS and whitefly feeding), although the magnitude of these effects may vary by cultivar. We confirmed that trichomes are associated to resistance to whiteflies in tomato, and found that trichome density is greater under DS, although this varies by type of trichome and cultivar. Thus, with the caveat of this variation, DS can prime plant resistance against whiteflies. In terms of tolerance of herbivory, we found lower compensatory ability under DS in three of the four cultivars studied, which is consistent with a resistance-tolerance trade-off. Our findings highlight the importance of considering both water availability and the nature and history of abundance of the most prevalent herbivores in a given agroecosystem before deciding on the cultivars or varieties to be planted. Without clear ways to predict these, the maintenance of genetic variation for resistance and tolerance against sap-sucking and chewing herbivores seems of utmost importance for food security.

## Figures and Tables

**Figure 1 plants-11-01049-f001:**
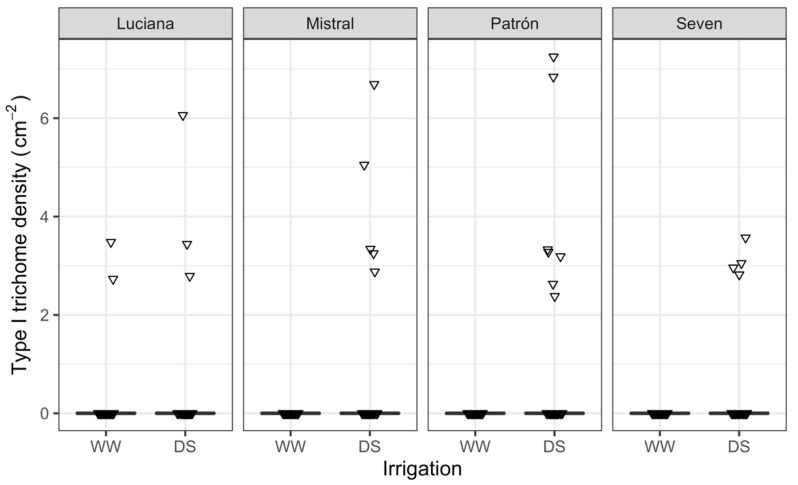
Boxplots of type I trichome density for plants of four tomato cultivars under contrasting irrigation treatments: well-watered (WW) and drought stress (DS). Boxplots appear as thick lines because all three first quartiles are zero in each case. Inverted triangles depict individual data points: the only 21 non-zero values are visible (horizontal jitter added).

**Figure 2 plants-11-01049-f002:**
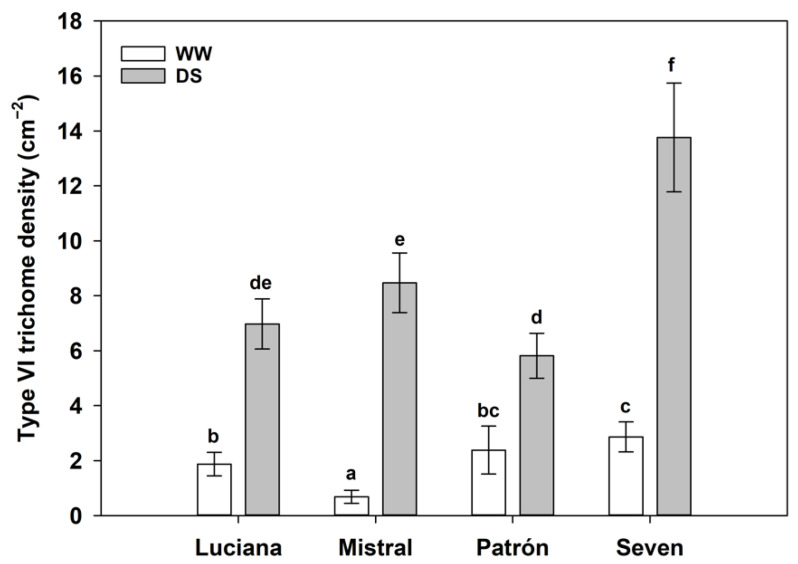
Type VI trichome density (estimated mean ± s.e.; back-transformed from ln) of plants from four tomato cultivars under contrasting irrigation treatments: well-watered (WW) and drought stress (DS). Letters above bars indicate results of Tukey–Kramer multiple comparisons: means of treatment combinations that share a letter are not significantly different at α = 0.05.

**Figure 3 plants-11-01049-f003:**
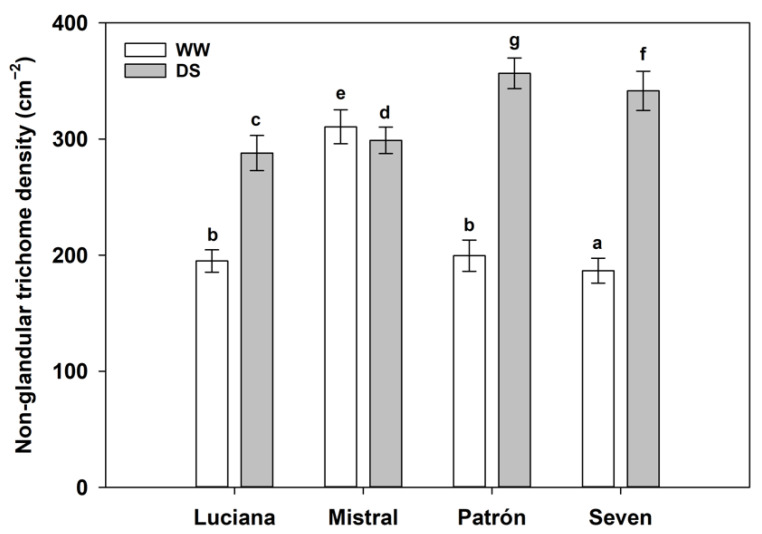
Non-glandular (Type V) trichome density (estimated mean ± s.e.; back-transformed from ln) of plants from four tomato cultivars under contrasting irrigation treatments: well-watered (WW) and drought stress (DS). Letters above bars as in Figure 2.

**Figure 4 plants-11-01049-f004:**
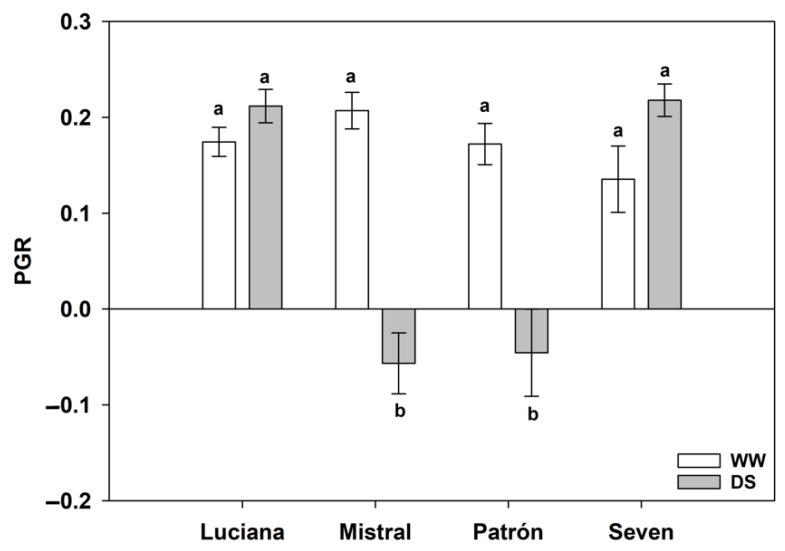
Population growth rate (PGR; estimated mean ± s.e.) of whitefly populations on plants of four tomato cultivars under contrasting irrigation treatments: well-watered (WW) and drought stress (DS). Letters above/below bars as in Figure 2.

**Figure 5 plants-11-01049-f005:**
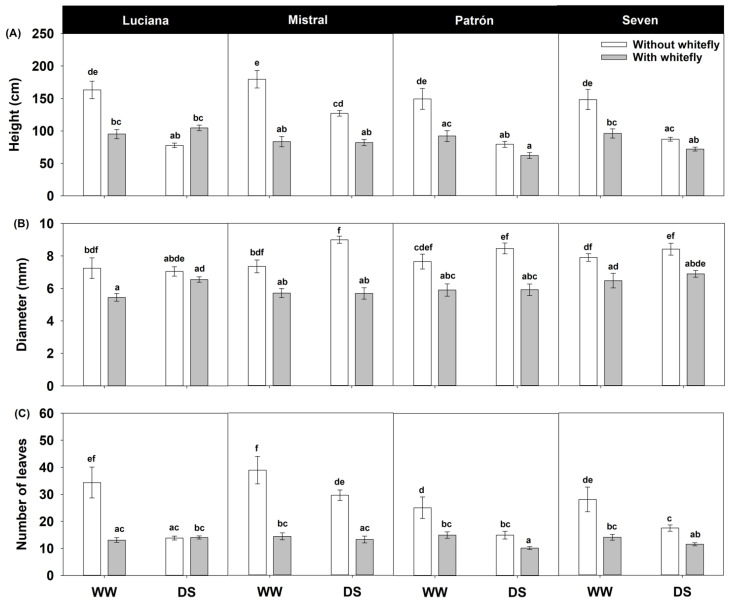
Height (**A**), diameter (**B**), and number of leaves (**C**) at harvest (estimated mean ± s.e.) of plants of four tomato cultivars under contrasting irrigation (well-watered, WW and drought stress, DS) and whitefly feeding treatments. Letters above bars as in Figure 2.

**Figure 6 plants-11-01049-f006:**
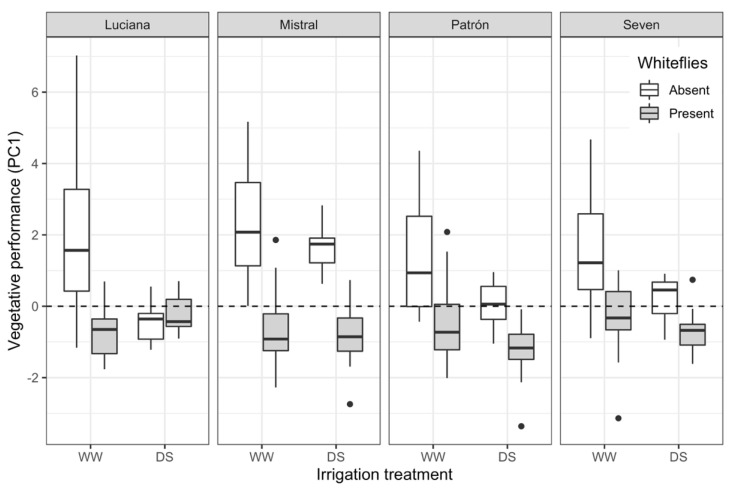
Boxplots of vegetative performance (Principal Component 1 from PCA analysis) of plants from four tomato cultivars under contrasting irrigation treatments: well-watered (WW) and drought stress (DS), with (shaded) and without (clear) whiteflies feeding on them. Each boxplot shows the interquartile range (box), the median (thick horizontal line within the box), “whiskers” that extend to the lowest or highest data value within a distance 1.5 times the length of the interquartile range from the lower or upper quartile, and individual values outside that range (dots).

**Figure 7 plants-11-01049-f007:**
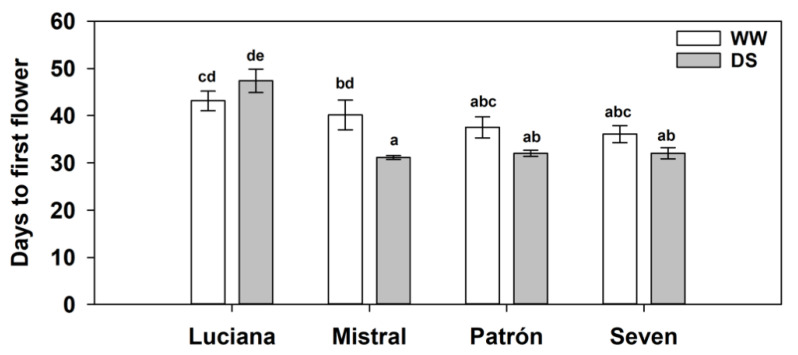
Days to first flower (estimated mean ± s.e.) of plants from four tomato cultivars grown under contrasting irrigation treatments: well-watered (WW) and drought stress (DS). Letters above bars as in Figure 2.

**Figure 8 plants-11-01049-f008:**
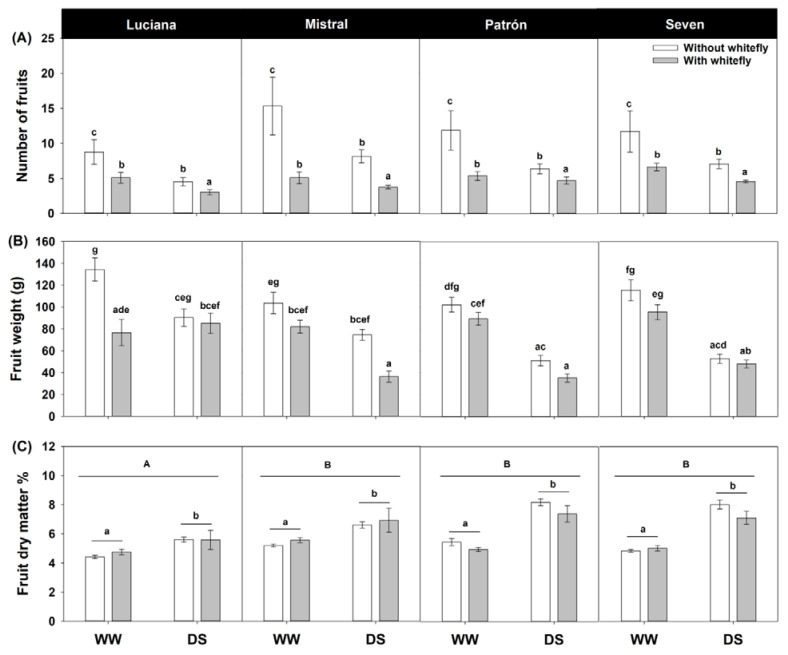
Estimated mean ± s.e of the number of fruits per plant (**A**), individual fruit fresh weight (**B**), and individual fruit dry matter content (**C**) of plants from four tomato cultivars grown under contrasting irrigation (well-watered, WW and drought stress, DS) and whitefly feeding treatments. Letters above bars as in Figure 2.

**Figure 9 plants-11-01049-f009:**
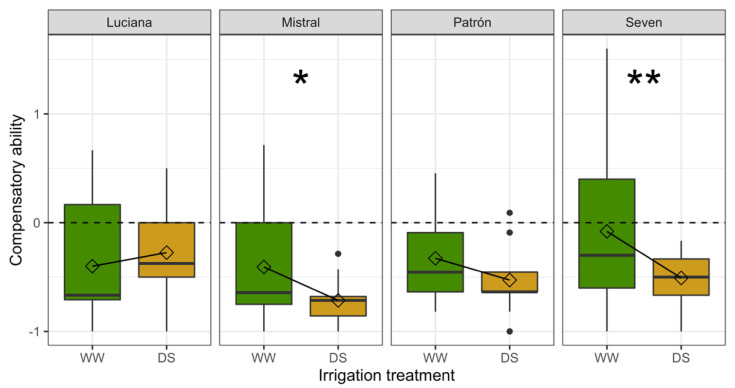
Boxplots of compensatory ability (CA) of plants after feeding damage by whiteflies. CA values less than zero indicate undercompensation (lower fitness of damaged plants compared to undamaged plants). Results of linear contrasts are indicated by asterisks: * *p* = 0.028; ** *p* = 0.002; lines join mean values within treatment combinations; other symbols as in Figure 6.

**Table 1 plants-11-01049-t001:** Summary statistics (degrees of freedom; *F* or *χ*^2^ statistic; *p* values) of the general linear model analyses for the effects of cultivar, irrigation, whitefly presence, and their interactions on plant height, diameter, and number of leaves produced.

		Height		Diameter		Number of Leaves	
	d.f.	*F*	*p*	*F*	*p*	*χ* ^2^	*p*
Cultivar	3	5.26	0.002	3.96	0.009	73.14	<0.0001
Irrigation	1	71.87	<0.0001	7.97	0.005	130.15	<0.0001
Whitefly	1	102.20	<0.0001	102.71	<0.0001	478.09	<0.0001
Cultivar × Irrigation	3	2.08	0.103	0.16	0.920	16.32	0.0010
Cultivar × Whitefly	3	7.40	<0.0001	3.01	0.031	37.72	<0.0001
Irrigation × Whitefly	1	51.06	<0.0001	0.65	0.419	42.21	<0.0001
Cultivar × Irrigation × Whitefly	3	2.95	0.033	3.19	0.024	31.42	<0.0001

## Data Availability

Data sets archived and publicly available at: https://doi.org/10.5683/SP3/NF5MXO (accessed on 21 February 2022).

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
