# Peer review of "Tomato Defense against Whiteflies under Drought Stress: Non-Additive Effects and Cultivar-Specific Responses"

_plants, 2022, doi:10.3390/plants11081049_

Round 1
Reviewer 1 Report
The author conducted a study to measure the combined effects of drought stress and whitefly herbivory in 4 different tomato cultivars. The physiological, molecular, and genetic interactions between plant responses to different types of stresses are not well understood and often show complex interactions. In this study the authors show that both abiotic and biotic stresses reduce the vegetative and reproductive traits of different tomato cultivars and provide compelling evidence that suggests that the combined effect of these stresses is another complex trait, that is primarily non-additive. The results also suggest that trichome density, which increases with drought stress, could be a factor that increases plant resistance to biotic stress and reduces the non-additive effect of both stresses.
I am curious as to why were these 4 tomato cultivars selected? Are there published studies of tomato drought- or whitefly-resistance across tomato genetic diversity and were any of these 4 lines included in such a study?
It wasn’t specifically stated, but I was wondering if during the whole duration of the experiment you kept track of individual plants for each type of measurement? Considering the hypothesis that trichome density could increase plant resistance to whiteflies, it could have been interesting to see the correlation plots (or some paired tests) between trichome density and whitefly resistance.
Some points of concern:
- I think the first introduction paragraph would benefit from a brief general discussion of the expected increases in biotic stress severity among all crops as climate change increases abiotic stresses.
- The introduction should include a paragraph about different types of trichomes, trichome density, and their role in abiotic and biotic stress resistance.
- Figure 1: Box-plots don’t seem to be the proper way to represent the data as it only shows outliers. With 459 out of 480 plants showing zero type I trichomes this figure might be replaced with a table and the statistics from Tabls S1.
- Sentence 493 “Type I trichome counts were generally zero (459 of 480 counts).“ contradicts sentence 106: “Type I glandular trichomes were present in all DS plants, regardless of the cultivar, albeit at very low densities.”.
- I like the way the results were summarized in figure 6 but the Y-axis label should not be “Vegetative performance (PC1)” as it is not proven or cited that plant height, diameter and number of leaves could be used as a proxy for vegetative performance. I think “Vegetative traits (PC1)” would be more appropriate.
- Chapter 4.1 Study System is a great paragraph that would work much better as part of the first introduction paragraph than as a chapter of methods.
- Line 510 “…DS can definitely prime plant”: better replace the word “definitely” for a less strong word statement.
- The descriptions in the figure legends should be more standardized. For example, some show if the data is mean ± S.E. and some mention correction for multiple testing (Tukey-Kramer) while others don’t.
Methods and their descriptions seem fine.
You don’t have to respond or address this comment: I have a feeling that moving the trichome results and discussion to the end of the results and discussion would create a more compelling story.
Author Response
Reviewer 1:
The author conducted a study to measure the combined effects of drought stress and whitefly herbivory in 4 different tomato cultivars. The physiological, molecular, and genetic interactions between plant responses to different types of stresses are not well understood and often show complex interactions. In this study the authors show that both abiotic and biotic stresses reduce the vegetative and reproductive traits of different tomato cultivars and provide compelling evidence that suggests that the combined effect of these stresses is another complex trait, that is primarily non-additive. The results also suggest that trichome density, which increases with drought stress, could be a factor that increases plant resistance to biotic stress and reduces the non-additive effect of both stresses.
I am curious as to why were these 4 tomato cultivars selected? Are there published studies of tomato drought- or whitefly-resistance across tomato genetic diversity and were any of these 4 lines included in such a study?
Response: We chose four cultivars that grow well in the region of study and that are recommended to farmers based on several traits, including their resistance to whiteflies and other herbivores. We have used some of these in other studies and confirmed that they vary for resistance to whiteflies or chewing herbivores (Ramachandran et al 2020, Mymko & Avila-Sakar 2019). We have added this information on L 354.
It wasn’t specifically stated, but I was wondering if during the whole duration of the experiment you kept track of individual plants for each type of measurement? Considering the hypothesis that trichome density could increase plant resistance to whiteflies, it could have been interesting to see the correlation plots (or some paired tests) between trichome density and whitefly resistance.
Response:
- We kept track of plants used for measurements of lifetime performance (irrigation-whitefly experiment) over the 121 days of the experiment. Other plants were only used to raise whiteflies, and others were used to measure water deficit. We have added that information to in the Experimental Design section: L 361, 404.
- This type of correlation was not carried out since Mimko and Ávila-Sakar (2019) and Zhang et al. (2020) reported this type of finding.
Some points of concern:
- I think the first introduction paragraph would benefit from a brief general discussion of the expected increases in biotic stress severity among all crops as climate change increases abiotic stresses.
Response: Agreed, we have added the following on L 40: "Drought and herbivory, two such components commonly faced by plants of many habitats, and there combined effects on plants are expected to become exacerbated by climate change [1-3]."
- The introduction should include a paragraph about different types of trichomes, trichome density, and their role in abiotic and biotic stress resistance.
Response: We have added that information on L 84: "In tomato (Solanum lycopersicum L.) and its close relatives, eight trichome types have been described [29]. Four of these are glandular (types I, IV, VI and VII) and four are non-glandular. These trichomes have been shown to confer resistance against whiteflies and chewing herbivores, and can also help plants to tolerate drought and cold [28,30,31]."
- Figure 1: Box-plots don’t seem to be the proper way to represent the data as it only shows outliers. With 459 out of 480 plants showing zero type I trichomes this figure might be replaced with a table and the statistics from Tabls S1.
Response: The 21 non-zero values appear as outliers because everything else is zero. We have provided a new plot where the outliers are jittered horizontally so that this is clear.
- Sentence 493 “Type I trichome counts were generally zero (459 of 480 counts).“ contradicts sentence 106: “Type I glandular trichomes were present in all DS plants, regardless of the cultivar, albeit at very low densities.”.
Response: Thank you for pointing this out. We have corrected the text in the results to read: "Only 21 plants had Type I glandular trichomes, and 19 of these were DS plants. In the WW treatment, only two Luciana plants had this kind of trichome (irrigation effect: ?² = 16.56, d.f. = 1, P < 0.0001; Figure 1)."
- I like the way the results were summarized in figure 6 but the Y-axis label should not be “Vegetative performance (PC1)” as it is not proven or cited that plant height, diameter and number of leaves could be used as a proxy for vegetative performance. I think “Vegetative traits (PC1)” would be more appropriate.
Response: One of the main reasons of conducting a PCA is precisely to come up with a measure that summarises plant vegetative performance, rather than showing only the separate (and correlated) traits measured. In the paragraph above the figure (L163), we stated that Principal Component 1 (PC1) accounted for 0.74 of the variance and had loadings of 0.61, 0.48, and 0.63. Given the large proportion of variance accounted by PC1, and the large positive loadings with the three variables, we followed by stating that we would use "PC1 as an overall measure of vegetative performance ...". We consider that this information allows the reader to understand the nuances with which we use the term.
- Chapter 4.1 Study System is a great paragraph that would work much better as part of the first introduction paragraph than as a chapter of methods.
Response: We moved the paragraph to the introduction.
- Line 510 “…DS can definitely prime plant”: better replace the word “definitely” for a less strong word statement.
Response: We deleted the word, as suggested.
- The descriptions in the figure legends should be more standardized. For example, some show if the data is mean ± S.E. and some mention correction for multiple testing (Tukey-Kramer) while others don’t.
Response: We added missing information for Figure 8. For brevity we refer readers to Figure 2 legend whenever T-K multiple comparisons were conducted.
Reviewer 2 Report
This manuscript is addressing an interesting topic: the combination impact of whitefly and drought stress on tomato plants. This is a novel approach because normally those aspects are evaluated separately.
This manuscript has all standards elements for a scientific article. Several variables including trichome density, whitefly performance, flowering, production and compensatory ability were evaluated and analyzed properly.
You should include some pictures showing damages caused by whitefly and drought stress and trichomes.
Good job!
Author Response
- Reviewer 2
This manuscript is addressing an interesting topic: the combination impact of whitefly and drought stress on tomato plants. This is a novel approach because normally those aspects are evaluated separately.
This manuscript has all standards elements for a scientific article. Several variables including trichome density, whitefly performance, flowering, production and compensatory ability were evaluated and analyzed properly.
You should include some pictures showing damages caused by whitefly and drought stress and trichomes.
Good job!
Response: Thank you.
Reviewer 3 Report
This is a review of the manuscript “Tomato defense against whiteflies under drought stress: non-additive effects and cultivar-specific responses” A clear introduction and results sections are degraded by flaws in the methods where I had trouble figuring out how the experiment was put together. Something as simple as how many replicates in the experiment doesn’t work out cleanly into the number of plants from the nursery. Missing details like the fertilizer and soil used to grow the plants makes the research unreproducible. It was also unclear how much time elapsed from whitefly infestation to taking plant measurements. The fortunate thing is that these critical problems are (or should be) easily fixed.
The authors made an unusual choice in selecting Type II sums of squares (Line 492). It is not clear why this is an unbalanced data set, or what treatments/replicates were lost. Much of the literature uses Type III. What are the critical features of Type II that benefit this research and how do you mitigate the problems with that choice? The Type II SS have issues if there are significant interaction terms.
The authors use a binomial model for analysis of Type I trichome counts. Based on the description, I wonder why the authors chose to convert to presence-absence rather than a zero inflated Poisson model. I am simply wondering why that choice. There were 21 plants with Type I trichomes.
Another choice is to calculate prevalence of trichomes using propCI(x = 2, m = 1, n = 60, ci.method = "CP") from the binGroup2 package in R. However, the estimates range from 0 with a 0 -> 0.0596 confidence interval to 0.1 with a 0.0375 -> 0.2051 confidence interval suggesting no significant difference. Given that under 5% of samples had these trichomes, it is hard to see relevance of this outcome except to note that there is genetic variability.
There are enough hypotheses for explaining the plant-insect interaction that any data will support one of them. The existing research is relevant to the discussion, but I don’t see how it helps. The treatments are all presence-absence dichotomies of effectively continuous variables.
Line 22) Known
Line 103) Might want to describe the different types of glandular trichomes, and/or provide a reference for the nomenclature. The results jump from type I to type VI glandular trichomes. Does tomato lack all other types? I am then faced with non-glandular trichomes as well, and there is at least a type V.
Line 103: Add a short paragraph (or add a couple of sentences to ending paragraph) describing the experiment. I have no clue what “Luciana plants” are in line 108. Organize the manuscript in the way you want me to read it. Don’t expect me to shuffle to the end to find the methods just to interpret the results section. Adapt writing to journal format. I google “Luciana plants” and discover that you might be working with Nymphaea ‘Luciana’ (a water lily) or Arctostaphylous luciana (a species of manzanita). Please do not make me guess, it doesn’t usually end well.
Line 107) do not use “very low densities” provide a number. The phrase does not carry any meaning because it lacks any context, especially at this point in the manuscript.
Figure 2) y-axis legend should be “Type VI trichome density (trichomes/cm2)” and similar for other figures as appropriate.
Figures 6,7,8) are there letters missing or wrong? Height (Patron) WW with whitefly is “ac” so it is significantly different from b, but not a or c. Diameter Luciana, DS whitefly is significantly different from b and c, but not a or d?
Figure 7) Why run a multiple comparison for all possible comparisons over all treatments if what you present is just the pairwise contrast within each cultivar?
Line 364) This is more introduction than methods.
Line 373) what do you mean by “whitefly-mediated diseases?” Sooty mold is not a disease. Sooty mold will colonize any sugary residue regardless of source: aphid, whitefly, sharpshooter, psyllid …
Line 387) vector is a better description of their role in the disease cycle.
Line 384) 257 of each cultivar, or 257 total? You cannot divide 257 into 4 equal portions. There is 1 plant left over.
Line 387) What kind of soil? Were the plants ever fertilized? Nutrients affect plant growth and host plant suitability, so this information is critical in reproducing your research.
Line 397) Cubicles contained two plants of each cultivar, or a cubicle contained two plants, and you had 178 cubiles (or 179 where one cubicle had only one plant).
Line 397) “Avoiding large discrepancies in the number of whiteflies” What were the observed differences in the number of whiteflies on each cultivar?
Water: 133 mm/day, 33 mm/day.
Line 403) drippers? I don’t understand droppers.
Line 417) you started with 257 plants. 20% of these were used to assess drought stress, leaving 205 or 206 plants. The whitefly test was a separate experiment using the same design. So five plants of each cultivar, leaving 186 plants for the main experiment.
Line 419) “different cultivar” is uninformative. It takes no more effort or journal space to specify which cultivar.
Line 423) You released 220 individuals into each chamber. But you only had two chambers, and one was for the no whitefly treatment.
Line 430) The correct citation defining glandular trichome types is something like Luckwill 1943 as cited in Wilkens et al. [36]. If you happen to have a copy of Luckwill 1943, then you can cite as normal.
Line 436) Not quite true. While Avery et al look at trichomes, they did not use or define your trichome type designations.
Line 438) Are these in addition to the 220 individuals, a part of the 220 individuals, or a separate experiment?
Seedlings were purchased from a grower, transplanted, and allowed to grow for 3 weeks. Drought stress treatments were then applied, and the plants allowed to “adapt” to their new conditions for how long? If they had 60 days to whitefly infestation, then 60 – 21= 39 days. Plus, ten days for the whitefly performance experiment, and unspecified for the main experiment. It does not seem like enough time for the plants to begin to senesce (line 458).
This paper by Ramachandran would suggest 6 months or more to senescence (This is the original Masters thesis that resulted in the paper that you cited): https://winnspace.uwinnipeg.ca/bitstream/handle/10680/1507/Ramachandran%2C%20Sreedevi.%20Soil%20N%20level%20and%20Tomato-whitefly%20interaction.pdf?sequence=1&isAllowed=y
Line 463) Statistical what?
My best guess at replication: There are 4 cultivars, 2 water treatments and 2 whitefly treatments for 8*2*2=32 treatments. 32 goes into 186 5 times, though it is close to 6 and I am not sure I have all the numbers right. So, 5 or 6 replicates per treatment?
Line 477) The compensatory ability of the ith individual is the ratio of the fruits produced by an individual subjected to herbivory divided by the median fruits produced by herbivore free individuals, minus one.
- I don’t understand why -1. Is there a real advantage to having the threshold at 0 versus 1? Is this more than just cosmetic?
- I don’t understand why this is restricted to DS plants.
- Is the compensatory ability of the WW plants different?
Line 492) The use of Type II SS is questionable. You have significant interaction terms.
Line 500) can remove “for figures” people familiar with R and tidyverse/ggplot2 will know, and it will not help people who are not familiar with data analysis in R.
Line 507) I am not convinced that you showed that trichome density is important in resistance/tolerance. Usually this is done by shaving trichomes so that the plant chemistry is more similar. In this study trichome density was manipulated by cultivar, with indications that the cultivars differ in other traits as well.
Table S1) Thank you for this table.
Table S1) n(non-missing) is useful, but the sample size that one calculates from the text should match the sample size here.
Table S1) Fruit fresh weight looks like it has the wrong n. My only guess is that individual fruits were used as replicates. You should start by averaging the fruits for each pot, and then do the analysis.
Table S1) Why is there a different n for fresh weight and dry weight?
Author Response
Reviewer 3
This is a review of the manuscript “Tomato defense against whiteflies under drought stress: non-additive effects and cultivar-specific responses” A clear introduction and results sections are degraded by flaws in the methods where I had trouble figuring out how the experiment was put together. Something as simple as how many replicates in the experiment doesn’t work out cleanly into the number of plants from the nursery.
Response: We have added information that should make the experimental design clearer. The specific lines changed are listed in response to the correspnding comments below.
Missing details like the fertilizer and soil used to grow the plants makes the research unreproducible. It was also unclear how much time elapsed from whitefly infestation to taking plant measurements. The fortunate thing is that these critical problems are (or should be) easily fixed.
Response: We have added that information in the Methods section.
The authors made an unusual choice in selecting Type II sums of squares (Line 492). It is not clear why this is an unbalanced data set, or what treatments/replicates were lost. Much of the literature uses Type III. What are the critical features of Type II that benefit this research and how do you mitigate the problems with that choice? The Type II SS have issues if there are significant interaction terms.
Response: The main reason for using Type II SS for some variables is that our data for those variables were unbalanced, which makes the effects of different explanatory variables non-orthogonal: the sequential SS calculated for a given variable depend on the position of that variable in the model (Hector et al. 2010). Type III SS have been criticized for testing for main effects in the presence of interactions (Hector et al. 2010, Langsrud 2003). In our study, second- or third-order interactions were significant, making less important which way we calculated SS, as the effect of one variable will depend on the level of the other variable(s). We have added the two references mentioned above in the manuscript.
The variables for which we are missing values have n=256. We have added a detailed explanation of the sample sizes in a new first section of Experimental design (L349).
The unbalanced data sets had the following number of observations:
variety Luciana Mistral Patrón Seven
whitefly Absent Present Absent Present Absent Present Absent Present
water
WW 13 20 12 20 10 20 11 20
DS 13 20 13 20 12 20 12 20
>The authors use a binomial model for analysis of Type I trichome counts. Based on the description, I wonder why the authors chose to convert to presence-absence rather than a zero inflated Poisson model. I am simply wondering why that choice. There were 21 plants with Type I trichomes.
Response: We considered different approaches, including a zero-inflated model. We decided there is little to gain from a zero-inflated model applied to data that are mostly zeros with some non-zero values (rather than more non-zero values than zeros). For the most part, a simple description of where the non-zero values occur suffices. The binomial model takes that description a step further, confirming a strong irrigation effect. As figure 1 shows, plants that had type I trichomes had them with densities between 2 and 6 trichomes per square cm., with full overlap among plants in different cultivar-by-irrigation level combination.
Another choice is to calculate prevalence of trichomes using propCI(x = 2, m = 1, n = 60, ci.method = "CP") from the binGroup2 package in R. However, the estimates range from 0 with a 0 -> 0.0596 confidence interval to 0.1 with a 0.0375 -> 0.2051 confidence interval suggesting no significant difference. Given that under 5% of samples had these trichomes, it is hard to see relevance of this outcome except to note that there is genetic variability.
Response: We maintain that there is more to gain by looking at the data on Figure 1 than trying complicated statistics that will likely not meet model assumptions. What we can see, and corroborate with the binomial model, is that 19 of the 21 plants with type 1 trichomes happen to be in the DS treatments. It is hard to explain this result as not related to the irrigation level.
There are enough hypotheses for explaining the plant-insect interaction that any data will support one of them. The existing research is relevant to the discussion, but I don’t see how it helps. The treatments are all presence-absence dichotomies of effectively continuous variables.
Response: We are unsure about the reviewer's point. Our results clearly do not support the Plant Stress Hypothesis. The fact that our data are consistent with two hypotheses says more about the non-mutually exclusive nature of the hypotheses than about our study. Lastly, as stated in our conclusion, drought stress primes tomato resistance to whiteflies, and we provide one more case of evidence of the resistance-tolerance trade-off. These are contributions to both basic and applied understanding of plant responses to combined biotic agents and abiotic factors.
Line 22) Known
Response: Corrected, thanks.
Line 103) Might want to describe the different types of glandular trichomes, and/or provide a reference for the nomenclature. The results jump from type I to type VI glandular trichomes. Does tomato lack all other types? I am then faced with non-glandular trichomes as well, and there is at least a type V.
Response: We have added this information you can find in the introduction L84:
"In tomato (Solanum lycopersicum L.) and its close relatives, eight trichome types have been described [29]. Four of these are glandular (types I, IV, VI and VII) and four are non-glandular. These trichomes have been shown to confer resistance against whiteflies and chewing herbivores, and can also help plants to tolerate drought and cold [28,30,31]."
Line 103: Add a short paragraph (or add a couple of sentences to ending paragraph) describing the experiment. I have no clue what “Luciana plants” are in line 108. Organize the manuscript in the way you want me to read it. Don’t expect me to shuffle to the end to find the methods just to interpret the results section. Adapt writing to journal format. I google “Luciana plants” and discover that you might be working with Nymphaea ‘Luciana’ (a water lily) or Arctostaphylous luciana (a species of manzanita). Please do not make me guess, it doesn’t usually end well.
Response: This particular journal format, with the results before the methods, does take adapting for both the authors and the readers. We have moved information about the study system onto the last part of the introduction to provide more context regarding the use of four tomato cultivars.
Line 107) do not use “very low densities” provide a number. The phrase does not carry any meaning because it lacks any context, especially at this point in the manuscript.
Response: We have removed the phrase.
MA - Armando: Figure 2) y-axis legend should be “Type VI trichome density (trichomes/cm2)” and similar for other figures as appropriate.
Response: Corrected.
Figures 6,7,8) are there letters missing or wrong? Height (Patron) WW with whitefly is “ac” so it is significantly different from b, but not a or c. Diameter Luciana, DS whitefly is significantly different from b and c, but not a or d?
Response: Letters are not missing. As stated in the legend of Fig 2: any two means that share a letters are not significantly different. Letters are assigned following the algorithm in Piepho 2004 Journal of Computational and Graphical Statistics, Volume 13, Number 2, Pages 456–466
Figure 7) Why run a multiple comparison for all possible comparisons over all treatments if what you present is just the pairwise contrast within each cultivar?
Response: Multiple comparisons allow us to make unplanned comparisons among any treatment combinations. Since we are also interested in the performance of different cultivars (indicative of genetic variation in the species) we have added: "Differences among cultivars were only significant under DS: Luciana took longer to flower than the other ones."
Line 364) This is more introduction than methods.
Response: We have moved this paragraph to the introduction.
Line 373) what do you mean by “whitefly-mediated diseases?” Sooty mold is not a disease. Sooty mold will colonize any sugary residue regardless of source: aphid, whitefly, sharpshooter, psyllid …
Response: Corrected.Line 387) vector is a better description of their role in the disease cycle.
Response: Changed accordingly
Line 384) 257 of each cultivar, or 257 total? You cannot divide 257 into 4 equal portions. There is 1 plant left over.
Response: For this an the comment for former L 397, we have added detailed information in the experimental design section:
"Initially, 480 seedlings were established (120 plants per cultivar); 120 plants were used to assess resistance based on whitefly population growth. Of the remaining 360 plants, we took 320 to split equally between whitefly treatments (160 with / 160 without). The rest were used to measure water potential (destructive sampling). However, we later realized we needed more plants for water potential measurements to be representative of the two irrigation treatments in each cultivar, so we used 56 plants without whiteflies for these measurements. This left us with 104 plants without whiteflies, eight of which died during the study. Thus, in total we had 160 plants with whiteflies (80 WW, 80 DS), and 96 without whiteflies (46 WW, 50 DS)."
Line 387) What kind of soil? Were the plants ever fertilized? Nutrients affect plant growth and host plant suitability, so this information is critical in reproducing your research.
Response: We have added the following on L 358: "Plants were not subjected to fertilization, to avoid potential changes in resistance due to mineral content [38]".
Line 397) Cubicles contained two plants of each cultivar, or a cubicle contained two plants, and you had 178 cubiles (or 179 where one cubicle had only one plant).
Response: We have added the following text on L 376: "Chambers were split into 16 cubicles: two per cultivar. Plants were assigned to cubicles according to their cultivar so as to avoid potential discrepancies in the number of whiteflies per plant due to differences in cultivar attractiveness to whiteflies (10 plants per cubicle for the chamber with whiteflies, 6 plants per cubicle for the chamber without whiteflies)."
Line 397) “Avoiding large discrepancies in the number of whiteflies” What were the observed differences in the number of whiteflies on each cultivar?
Response: We placed 220 whiteflies per cubicle, which corresponds to 22 whiteflies / plant. The cubicles precluded whiteflies moving among cultivars. We changed the text to make it clearer (L401).
Water: 133 mm/day, 33 mm/day.
Response: Corrected: 133 ml/day, 33 ml/day.
Line 403) drippers? I don’t understand droppers.
Response: Corrected.
Line 417) you started with 257 plants. 20% of these were used to assess drought stress, leaving 205 or 206 plants. The whitefly test was a separate experiment using the same design. So five plants of each cultivar, leaving 186 plants for the main experiment.
Response: This is now explained on the Experimental Design section, as mentioned above.
Line 419) “different cultivar” is uninformative. It takes no more effort or journal space to specify which cultivar.
Response: Changed accordingly.
Line 423) You released 220 individuals into each chamber. But you only had two chambers, and one was for the no whitefly treatment.
Response: We meant cubicle. Thanks for noticing the error. Changed accordingly
Line 430) The correct citation defining glandular trichome types is something like Luckwill 1943 as cited in Wilkens et al. [36]. If you happen to have a copy of Luckwill 1943, then you can cite as normal.
Response: Thank you, we had the references in the wrong place in the sentence. These references refer to the putative resistance function of trichomes. We changed the text as follows (L 408): "To assess the influence of DS on trichome density as a putative mechanism of resistance [28,51,62], we identified and counted glandular (types I and VI) and non-glandular (type V) trichomes on three 1-cm2 pieces taken from each of two leaves per plant-cultivar-irrigation treatment."
JV: Line 436) Not quite true. While Avery et al look at trichomes, they did not use or define your trichome type designations.
Response: Thank you. We had the wrong reference. We changed the text in L 415 to " Trichomes were classified, counted, and recorded as per McDowell et al. [29]."
Line 438) Are these in addition to the 220 individuals, a part of the 220 individuals, or a separate experiment?
Response: This is now explained on the Experimental Design section, as mentioned above.
Seedlings were purchased from a grower, transplanted, and allowed to grow for 3 weeks. Drought stress treatments were then applied, and the plants allowed to “adapt” to their new conditions for how long? If they had 60 days to whitefly infestation, then 60 – 21= 39 days. Plus, ten days for the whitefly performance experiment, and unspecified for the main experiment. It does not seem like enough time for the plants to begin to senesce (line 458).
This paper by Ramachandran would suggest 6 months or more to senescence (This is the original Masters thesis that resulted in the paper that you cited): https://winnspace.uwinnipeg.ca/bitstream/handle/10680/1507/Ramachandran%2C%20Sreedevi.%20Soil%20N%20level%20and%20Tomato-whitefly%20interaction.pdf?sequence=1&isAllowed=y
Response: No change needed. We did state that mature fruits had been harvested 107-121 days after transplant (L 434)
Line 463) Statistical what?
Response: L 438 We added the missing word: "analyses".
My best guess at replication: There are 4 cultivars, 2 water treatments and 2 whitefly treatments for 8*2*2=32 treatments. 32 goes into 186 5 times, though it is close to 6 and I am not sure I have all the numbers right. So, 5 or 6 replicates per treatment?
Response: This is now explained on the Experimental Design section, as mentioned above.
Line 477) The compensatory ability of the ith individual is the ratio of the fruits produced by an individual subjected to herbivory divided by the median fruits produced by herbivore free individuals, minus one.
- I don’t understand why -1. Is there a real advantage to having the threshold at 0 versus 1? Is this more than just cosmetic?
- I don’t understand why this is restricted to DS plants.
- Is the compensatory ability of the WW plants different?
Response: We mistakenly wrote "DS" instead of "irrigation". We have corrected that. Compensatory ability was calculated similarly for WW and DS plants. The reference to 0 as indicative of equal-compensation is a convention (see also Strauss and Agrawal 1999). It makes sense to us that no detrimental or positive effect of damage be scored as zero (no change in fitness is a change of zero).
Line 492) The use of Type II SS is questionable. You have significant interaction terms.
Response: We have responded above to this question.
Line 500) can remove “for figures” people familiar with R and tidyverse/ggplot2 will know, and it will not help people who are not familiar with data analysis in R.
Response: Removed as suggested
Line 507) I am not convinced that you showed that trichome density is important in resistance/tolerance. Usually this is done by shaving trichomes so that the plant chemistry is more similar. In this study trichome density was manipulated by cultivar, with indications that the cultivars differ in other traits as well.
Response: Shaving has its own set of problems, starting with the damage caused to the leaf and the induction of metabolic pathways that can have consequences on a variety of responses, including resistance, compensation and growth. We have changed the wording of L 475 to the more conservative: "We confirmed that trichomes are associated to resistance to whiteflies in tomato, and found that trichome density is greater under DS, although this varies by type of trichome and cultivar."
Table S1) Thank you for this table.
Table S1) n(non-missing) is useful, but the sample size that one calculates from the text should match the sample size here.
Response: Information on sample sizes has been added above.
Table S1) Fruit fresh weight looks like it has the wrong n. My only guess is that individual fruits were used as replicates. You should start by averaging the fruits for each pot, and then do the analysis.
Response: Thank you for noticing this. We have now included the correct analyses, using the averages of fresh or dry weights, as suggested. Results of fruit fresh weight did not change importantly, but no was significant for dry matter. We changed the text accordingly: "Fruit fresh weight also tended to be negatively affected by whitefly feeding, but the magnitude and significance of the effect depended on both cultivar and irrigation (cultivar-irrigation-whitefly interaction: F 3, 192 = 3.98, P = 0.009; Figure 8B). Under WW conditions, the negative effect of whitefly feeding on fruit fresh weight was significant only in Luciana, and under DS, the same was true only in Mistral. Indeed, in Mistral, the fruits with the lowest fresh weights were produced under combined DS and whitefly feeding. DS alone also had a negative effect on fruit fresh weight, but this was only marginally significant in Luciana. In Seven, fruit fresh weight was greater on WW plants regardless of whitefly presence (Figure 8B). Fruit dry matter content was generally greater in DS than WW plants (F 1,192 = 62.71; P < 0.0001; Figure 8C). Fruits of Luciana had lower dry matter content than those of the other cultivars (F 3,192 = 3.96; P = 0.009)."
Table S1) Why is there a different n for fresh weight and dry weight? -
Response: This was an error that no longer applies with the correct analyses.